# Real-Time Detection of Bud Degeneration in Oil Palms Using an Unmanned Aerial Vehicle

**Alexis Vázquez-Ramírez** [1], **Dante Mújica-Vargas** [1,*], **Antonio Luna-Álvarez** [1], **Manuel Matuz-Cruz** [2] and **José de Jesus Rubio** [3]

1. Department of Computer Science, Centro Nacional de Investigación y Desarrollo Tecnológico, Interior Internado Palmira S/N, Palmira, Cuernavaca 62490, Mexico; alexxis.97@outlook.com (A.V.-R.); d18ce009@cenidet.tecnm.mx (A.L.-Á)
2. Department of Computer Science, Instituto Tecnológico De Tapachula, Tapachula 30700, Mexico; mjmatuz@tapachula.tecnm.mx
3. Escuela Superior de Ingeniería Mecánica y Eléctrica Unidad Azcapotzalco, Instituto Politécnico Nacional, Ciudad de México 02550, Mexico; rubio.josedejesus@gmail.com
* Correspondence: dante.mv@cenidet.tecnm.mx

**Abstract:** This paper presents a novel methodology for the early detection of oil palm bud degeneration based on computer vision. The proposed system uses the YOLO algorithm to detect diseased plants within the bud by analyzing images captured by a drone within the crop. Our system uses a drone equipped with a Jetson Nano embedded system to obtain complete images of crops with a 75% reduction in time and with 40% more accuracy compared to the traditional method. As a result, our system achieves a precision of 92% and a recall of 96%, indicating a high detection rate and a low false-positive rate. In real-time detection, the system is able to effectively detect diseased plants by monitoring an entire hectare of crops in 25 min. The system is also able to detect diseased plants other than those it was trained on with 43% precision. These results suggest that our methodology provides an effective and reliable means of early detection of bud degeneration in oil palm crops, which can prevent the spread of pests and improve crop production.

**Keywords:** bud degeneration; YOLO algorithm; unmanned aerial vehicles; embedded systems; oil palm

## 1. Introduction

Bud degeneration (BD) consists of poor growth and malformation of new leaves on the oil palm plant and is generally caused by factors such as leaf curling and/or bud rot [1,2]. In [3] , it was reported that the latter is the most devastating plague in oil palm crops in Latin America, as it affects production, nursery and prenursery plants. This disease consists of the deterioration of the immature tissues of the leaves and fruits of the plant, causing low oil production [4].

For crop monitoring, methods based on unmanned aerial vehicles and deep learning have been proposed, such as in [5,6]. Some works, such as [7–9], propose the use of neural networks for object detection within crops.As for proposals based on object detection models have been made in proposals based on real-time detection using deep learning and unmanned aerial vehicles yielding feasible results [10,11]. Some work shows that the use of drones for crop monitoring can improve the efficiency of agriculture by detecting diseases early and optimizing the use of resources such as water and agrochemicals [12]. Several authors propose the use of the YOLO algorithm to detect and localize objects in images, demonstrating the robustness of the YOLO algorithm [13,14]; however, there are no studies that have implemented these methods in the detection of diseases in oil palm crops, so this study aims to present an innovative and accurate methodology for the management of BD.

Currently, the methodology used to detect BD has shortcomings in terms of time and accuracy due to the rudimentary techniques implemented. The literature shows that there

is no formal technique to detect the disease, especially in the early stages of its growth. Palm oil companies such as Zitihualt S.P.R. de R.L. [15], in order to detect the occurrence of BD in their crops, make a tour of the entire crop looking for indicators such as dry or deformed leaves and plants with abnormal bud growth. These indicators proposed by [16] are located in the center of the plant and very often several meters above the ground, which makes them difficult to visualize. The complexity of the process lies in the subjective interpretation, which is given at the level of incidence on the plant, which could cause mistakes and obtain inconsistent and varied results [16,17]. For an early diagnosis, it is necessary to recognize the first symptoms by evaluating the leaves since the degree of severity depends on the sanitary measures to be applied [17]. CENIPALMA [18] recommends treating plants with BD only when there is a low degree of severity, allowing individual treatment of affected plants and their neighbors [1,19]. In this work, we present a novel methodology based on artificial vision and unmanned aerial vehicle (UAV) for real-time detection and monitoring of oil palm crops with BD incidence. The literature shows that the application of a UAV, involving a drone and convolutional neural network (CNN), provides a great advantage in time and accuracy with respect to the techniques currently used in crop monitoring [20,21]. The proposed system allows for work to be done with the sequences of images acquired during flight combined with the global positioning system (GPS) information to localize a diseased plant within a crop. A plant with BD has its own characteristics that distinguish it from a healthy plant, such as the color and shape of the leaves [2]; the application of an artificial neural network for pattern recognition makes it possible to identify these characteristics accurately. Our model allows for the early detection of the disease, which allows for better management and control of the affected plant.

The paper is organized as follows: Section 2 describes the proposed methodology, as well as the architecture of the object detection algorithm used. Section 3 details the experiments performed, as well as their validation and comparative analysis with the currently used methodology. Finally, Section 4 contains a discussion of the results of the work and observations on other applications.

## 2. Materials and Methods

This paper provides a methodology for real-time detection of plants with BD. The proposed system is able to work in complicated environments, allowing to reduce the response time to new pest outbreaks. The methodology is based on CNN and artificial vision for data collection using a drone. Figure 1 describes the methodology implemented to develop the proposed system.

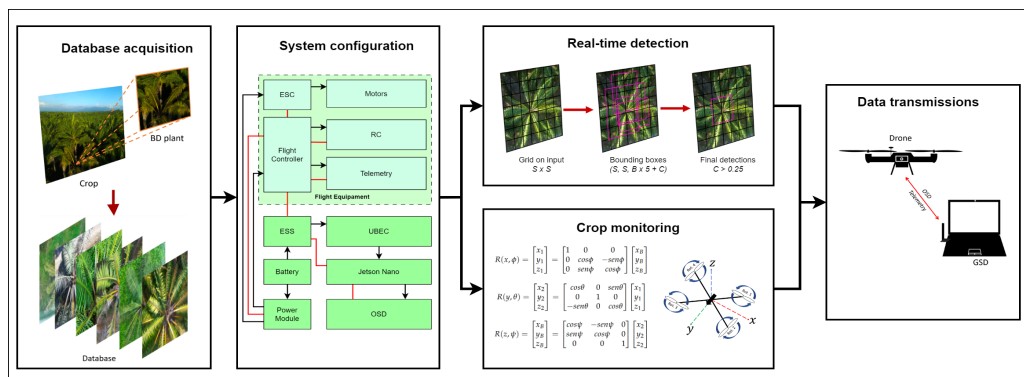

**Figure 1.** Proposed methodology. This was divided into 5 processes: the first was database acquisition, which consisted of collecting images of plants with BD; the second was system configuration according to the feeding diagram; the third and fourth were real-time detection and crop monitoring, which were performed in parallel during the detection of BD in the crop; the fifth was data transmission, which consisted of sending the real-time video and the status of the drone to the GSD. The first process was performed only for the training phase of the system, while the others were performed each time the crop was monitored.

We used an 8 megapixel camera for object detection with the you only look once (YOLO) algorithm [22] and the Nvidia Jetson Nano (JN) development kit [23] mounted on a UAV. The UAV (drone) had a full view of the crop and used telemetry to take a tour of the crop, looking for diseased plants and sending the location to the ground station Device (GSD).

## 2.1. Database Acquisition

The database design required a field study to obtain photographs of plants with BD. This consisted of collecting photographs of oil palm plants according to two criteria: plants with a BD incidence of 2 or more according to the CENIPALMA severity scale in Figure 2, and healthy plants, both between 4 and 22 years old. Given the location of the object of interest, a drone was used to photograph the bud of the plant, with photographs being taken at different angles and focal distances and at different heights above the bud of the plant in order to have a set of data with different environments. To take into account other factors such as climate (winds over 30 km/h, rain), the life cycle of the plant (beginning of reproduction and leaf growth), and the luminosity of the environment (sunny, cloudy, and sunset), photographs were taken over a period of 6 months.

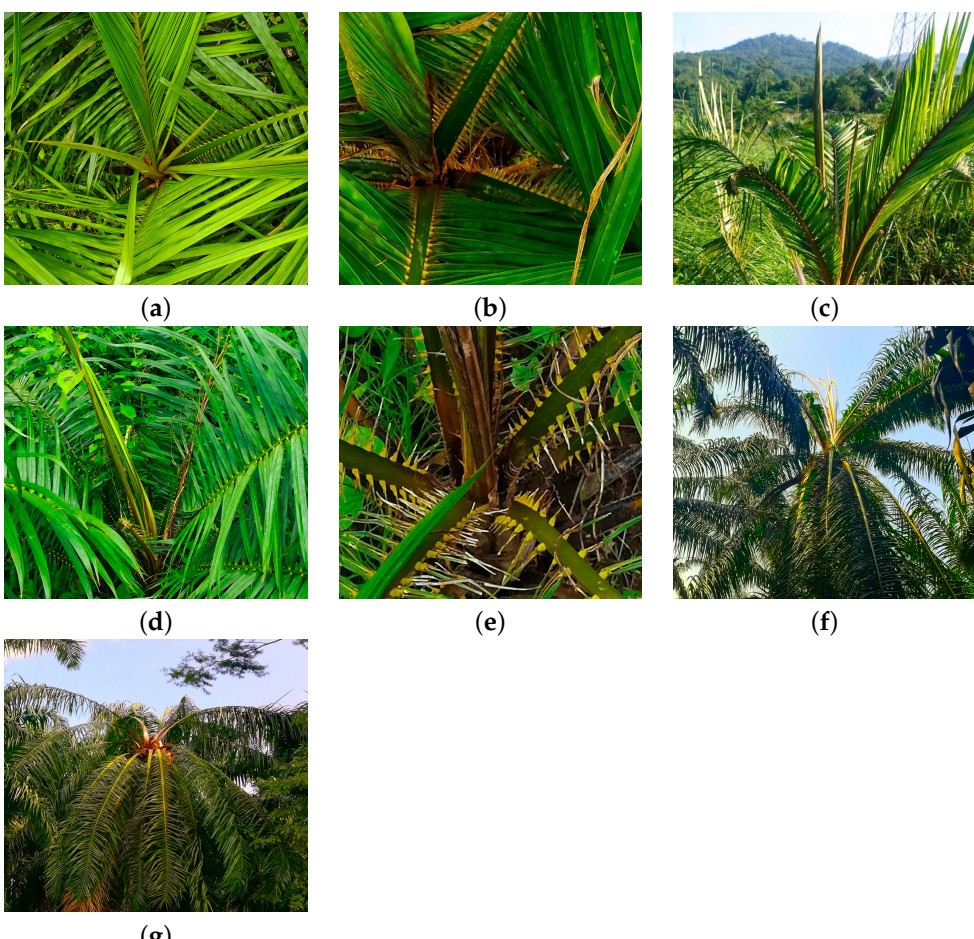

**Figure 2.** Severity scale for oil palm bud rot proposed by CENIPALMA [24,25]. (**a**) Grade 0, leaf shows vigor and health with no lesions; (**b**) grade 1, lesions cover 0.1 to 20% of the leaf area; (**c**) grade 2, lesions cover 20.1 to 40% of leaf area; (**d**) grade 3, lesions cover 40.1 to 60% of the shaft area; (**e**) grade 4, lesions cover 60.1 to 80% of the leaf area; (**f**) grade 5; lesions cover 80.1 to 100% of the leaf area; (**g**) grade 6, lesions cover 80.1 to 100% of the leaf area.

### 2.2. System Configuration

This phase describes the conditioning of the prototype, the configuration of the drone, and the JN Development Kit for the route it took over the crop in search of diseased plants. This process mainly consisted of checking the status of the drone's motors, the direction of rotation and propeller support, the battery status, the connection to the GSD, and the power supply connections. The full system diagram is shown in Figure 3.

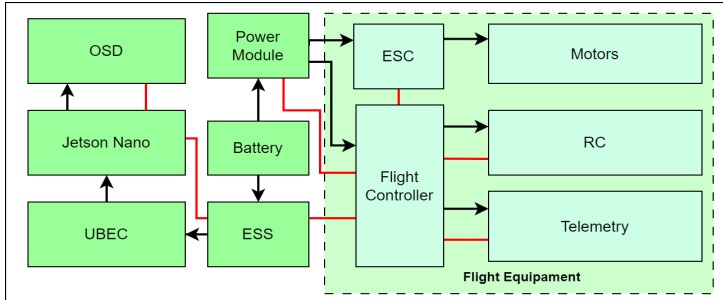

**Figure 3.** System power supply diagram. The Flight Equipment section shows the UAV components connected to JN via the ESS. The black arrows indicate the electrical power connections coming from the battery, and the red lines indicate the communication between the different components.

For flying over the crop, an important factor to consider is the flight time provided by the drone's battery, as it powers both the drone and the JN, so it is important to reduce the system's power consumption in case the connection to the remote control (RC) is lost or an emergency landing is required. Factors such as wind speed and crop expansion make it difficult to land the drone, so an emergency shutdown system (ESS) is implemented to prioritize power consumption only to the flight equipment. For the ESS design, we used a Digispark microcontroller to receive data from the $I2C - 2$ port of the flight controller (FC), which sends an electrical pulse to the JN, $J40$ port and cuts off the power through the voltage relay connected to the UBEC (undervoltage battery eliminator circuit) of the negative wire of the JN battery.

### 2.3. Crop Monitoring

In order to perform real-time detection of BD-infected plants within the crop, it is necessary to create a flight plan. To create the flight plan over the crop, we used Mission Planner software, which allowed us to set reference points that the drone could use as routes to navigate through the crop using a GPS receiver. The flight path of the drone was made over two rows of plants if the crop was less than 10 years old, and one row if it was older to get a better field of view of the plants. Figure 4 shows the traced flight path in an oil palm crop.

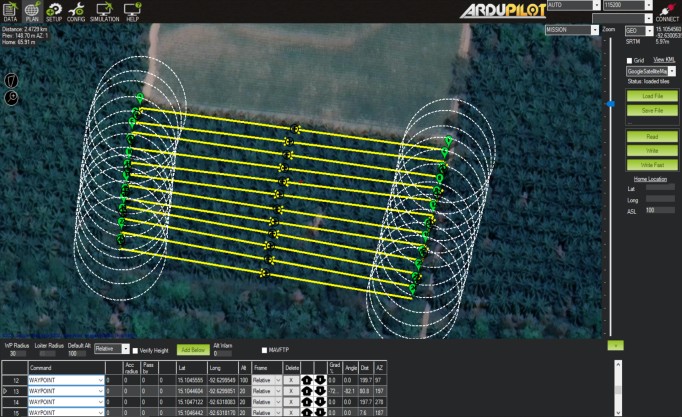

**Figure 4.** Palm crop flight plan created with Mission Planner software. The yellow lines indicate the UAV flight path and the green dots the waypoints.

The movement of the drone over the crop was based on rotation and translation, as determined by 3 fixed axes $(x, y, z)$ through the oscillation of the navigation angles: yaw (angle-$z$), pitch (angle-$y$), and roll (angle-$x$). This provided 6 degrees of freedom: three translational and three rotational— roll, pitch, and yaw. According to [26], the position of the drone is always defined by two points: one on the ground $E = xE, yE, zE$ and one at the center of mass of the drone $B = xB, yB, zB$. In a reference system $B$ as in Figure 5, the dynamic equations of rotation and translation are used, where this point is located at the center of the drone, defining the rotations and movements to be performed.

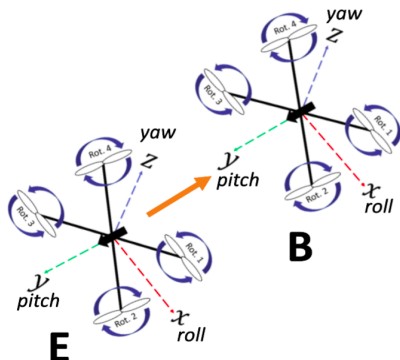

**Figure 5.** Reference system for drone motion. The UAV moves over time, changing its position with respect to a reference point E to B, which is assumed to be fixed along its three axes.

In the Figure 5, the position $\xi$ of the drone at point $E$ is defined by the vector $\xi = [x, z]^T$ and its orientation by the vector $\eta = [\phi \theta \psi]$. The orientation of the drone with respect to the reference frame $E$ is formulated as a function of the angles $[Z_\phi][Y_\theta][X_\psi]$. For the rotation matrix, the motion around the axes $[Z_\phi][Y_\theta][X_\psi]$ is used. Equation (1) defines the roll matrix, Equation (2) corresponds to the pitch matrix, and Equation (3) defines the yaw matrix [26–28].

$$R(x, \phi) = \begin{bmatrix} x_1 \\ y_1 \\ z_1 \end{bmatrix} = \begin{bmatrix} 1 & 0 & 0 \\ 0 & cos\phi & -sen\phi \\ 0 & sen\phi & cos\phi \end{bmatrix} \begin{bmatrix} x_B \\ y_B \\ z_B \end{bmatrix} \tag{1}$$

$$R(y, \theta) = \begin{bmatrix} x_2 \\ y_2 \\ z_2 \end{bmatrix} = \begin{bmatrix} cos\theta & 0 & sen\theta \\ 0 & 1 & 0 \\ -sen\theta & 0 & cos\theta \end{bmatrix} \begin{bmatrix} x_1 \\ y_1 \\ z_1 \end{bmatrix} \tag{2}$$

$$R(z, \psi) = \begin{bmatrix} x_B \\ y_B \\ z_B \end{bmatrix} = \begin{bmatrix} cos\psi & -sen\psi & 0 \\ sen\psi & cos\psi & 0 \\ 0 & 0 & 1 \end{bmatrix} \begin{bmatrix} x_2 \\ y_2 \\ z_2 \end{bmatrix} \tag{3}$$

The complete rotation matrix is calculated from the rotation matrices of Equations (1)–(3) as shown in Equation (4).

$$R(\phi, \theta, \psi) = R(z, \psi)R(y, \theta)R(x\phi) =$$

$$\begin{bmatrix} c\theta c\psi & c\psi s\theta s\phi - s\psi c\theta & c\psi s\theta c\phi + s\psi s\phi \\ s\psi c\theta & s\psi s\theta s\phi + c\psi c\theta & s\psi s\theta c\phi - sc\phi c\psi \\ -s\theta & c\theta s\phi & c\theta c\phi \end{bmatrix} \tag{4}$$

Equation (4) expresses the translational motion of the drone and is reduced when $\theta \to 0$ and $\phi \to 0$, where $cos\phi \cong 1$, $cos\theta \cong 1$, and $sen\phi \cong 0$, $sen\phi \cong 0$ are valid when the roll and pitch rotations are low. Equation (5) shows this simplified kinematic model.

$$\begin{bmatrix} \dot{x}_E \\ \dot{x}_E \\ \dot{x}_E \end{bmatrix} = \begin{bmatrix} cos\psi & -sen\psi & 0 \\ sen\psi & cos\psi & 0 \\ 0 & 0 & 1 \end{bmatrix} \cdot \begin{bmatrix} V_x \\ V_y \\ V_z \end{bmatrix} \tag{5}$$

The vector $[V_x V_y V_z] = V^T$ represents the drone velocities in the reference frame $B$ in Figure 5, and the vector $\xi = [\dot{x}\dot{y}\dot{z}]^T$ represents the drone velocities in the $x, y, z$ axes, respectively. To obtain the velocities at the point of inertia, the product $\xi = R(\phi, \theta, \psi)V$ is performed [26,28].

*2.4. Real-Time Detection*

For BD detection in oil palm, YOLOv3 was used due to its lower computational complexity compared to later versions, making it suitable for low-power devices such as JN. In addition, YOLOv3 has been shown to be effective in detecting objects in low-light situations and with small objects, making it suitable for BD detection in oil palm. It has also been optimized to detect multiple objects in a single image, which is useful for detecting multiple diseased plants in a single image.

For training, YOLO was adapted to run on JN. This was done using a transfer learning technique with pretrained weights provided by the author, and adjustments were made to the yolov3.cfg model configuration to reduce the input image size to $320 \times 320$; moreover, the *batch* and *subdivisions* parameters were set to 1, the *step* size was changedto 8000-9000, and the *max_batches* parameter was set to 10,000. In addition, parameters in the *makefile* were enabled to take advantage of the GPU power of the Jetson Nano. During training, the image input is taken and divided into a $S \times S$ grid. If the center of the object, in this case the plant with BD, is in a cell, that cell is responsible for detecting that object. Each cell predicts bounding box $B$ and *confidence* values for that box. Formally, *confidence* is defined as Equation (6).

$$Pr(Objet) \times IOU_{pred}^{truth} \tag{6}$$

where, if there is no object in that cell, the *confidence* score must be 0; otherwise the *confidence* score should be be equal to the intersection over union (IoU) between the predicted box and the defined box. Each bounding box consists of 5 predictions: $x, y, w, h$, and *confidence*. The variables $x$ and $y$ represent the center of the box relative to the grid cell boundaries, and $w, h$ represents the width and height relative to the full image. Each grid cell also predicts the $C$ probabilities of conditional class, $Pr(Class_i|Objet)$. These probabilities depend only on the cell containing the object. Furthermore, only the set class probabilities per cell are predicted, independent of the number of $B$. As shown in the Equation (7), YOLO multiplies the $C$ probabilities and the *confidence* values of each cell during validation. This gives class-specific *confidence* scores for each bounding box. These scores define the probability of the class appearing in the box and the probability of the box coinciding with the object [22,29].

$$Pr(Class_i Objet) \times Pr(Objet) \times IOU_{pred}^{truth}$$
$$= Pr(Class_i) \times IOU_{pred}^{truth} \tag{7}$$

For detection, the vector of predictions $(x, y, w, h, confidence)$ is defined such that each grid cell predicts $B$ bounding boxes and $c$ class probabilities. Where $x, y, w$, and $h$ are normalized between $[0, 1]$, relative to the image size. This prediction can be defined with Equation (8).

$$(S, S, B \times 5 + C) \tag{8}$$

For example, for a $448 \times 448$ image with a three-channel depth (RGB) and a probability of class $C = 1$, the prediction shape would be Equation (9).

$$(S, S, B \times 5 + C) = (7, 7, 2 \times 5 + 1) = (7, 7, 11) \tag{9}$$

where YOLO predicts a tensor $(7, 7, 11)$, reduces its spatial dimension to $7 \times 74$ with 1024 output filters at each location, takes an image, performs linear regression using two full connect layers, and makes $7 \times 7 \times 2$ bounding box predictions to make the final prediction, keeping only those bounding boxes with *confidence* values $(< 0.25)$ as the final detection, as seen in Figure 6. The architecture is based on 24 convolutional layers followed by two fully connected layers. The convolutional layers alternate to use a $1 \times 1$ reduction as an alternative when reducing the depth of the feature map. For the last convolutional layer, YOLO generates a tensor of shape $(7, 7, 1024)$ and then flattens the tensor. Using two fully connected layers as a linear regression shape, it generates parameters of $(7 \times 7 \times 11)$ and then goes to shape $(7, 7, 11)$, i.e., 2 bounding box predictions per location [22].

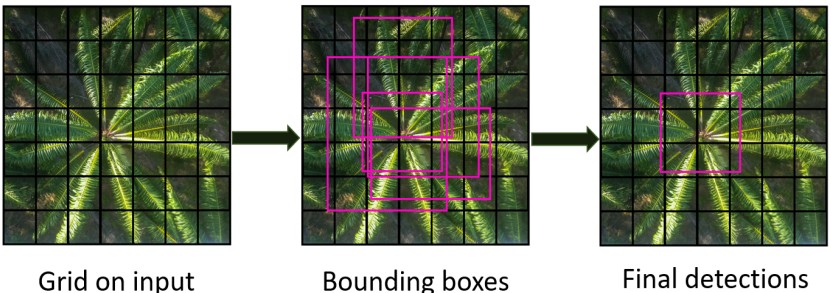

Grid on input          Bounding boxes          Final detections

**Figure 6.** YOLO detection process diagram. YOLO takes the input image, divides it into a $7 \times 7$ grid, predicts bounding boxes, and, according to a threshold, keeps only the boxes that contain the object.

### 2.5. Data Transmissions

For real-time positioning, our system uses a GPS receptor connected to the FC. This information is sent to the GSD via two telemetry receivers that are directly connected to avoid any kind of interference or disconnection. The real-time video connection is made via an On Screen Display (OSD) transmitter connected to the HDMI port of the JN and received by an HDMI display connected to an OSD receiver on the GSD.

Based on the triangulation method, GPS allows for determination of the real-time position of the drone in space from the distance to three other points whose coordinates are known. Suppose a point $B = (x, y, z)$ is located in space with unknown coordinates. Similarly, let $S_1, S_2, S_3$ be three fixed points in space whose coordinates are known and are $d_i = d(b, S_i), i = 1, 2, 3$. Since this is the distance from the unknown point to the three hypothetically known fixed points, it is necessary to determine $(x, y, z)$. If three spheres with center $S_i$ and radius $R_i$ are drawn, it is obvious that they intersect in at least one point, and it is obvious that this point, common to the three circles, will be $B$. That is, if the coordinates of the known points are $S_i = (x_i, y_i, z_i)$, we have three equations of circles defined as follows [30]:

$$(x - x_i)^2 + (y - y_i)^2 + (z - z_i)^2 = d_i^2 \tag{10}$$

for $i = 1, 2, 3$, or in terms of the standard, $\| B - S_i \|_2^2 = d_i^2$. They form the results solution of the nonlinear system, at least formally, in the coordinates of the point $B$ to be found, as shown in Figure 7.

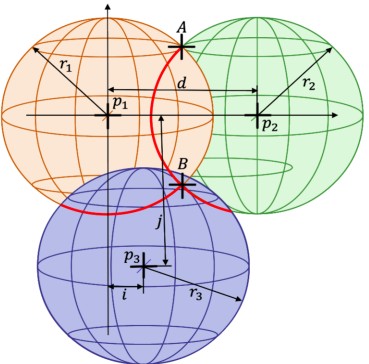

**Figure 7.** Positioning of point *B* in space. *B* and *A* is the position of the drone's GPS receiver over time.

Once point *B* is reached, which in this case would be the GPS receiver of the FC, the system sends this position, as well as its status in terms of battery level and the intensity of the connection with the RC, via telemetry to the GSD. For real-time video of the crop and any events that occur, the system sends the data through an OSD transmitter that is received on an HDMI display connected to an OSD receiver.

After the information from the drone is received in the ground station device, the incidents found are recorded for subsequent management by the farmer. The report includes the GPS location of the affected plant and its degree of severity.

## 3. Experimental Results

In order to evaluate the performance of the proposed system, a series of experiments were carried out, consisting of tuning the YOLO detection threshold, validation with the Test dataset, real-time detection of diseased plants within a crop, comparison with the currently used methodology, and validation with images of plants in completely different environments from those in which the model was trained. To quantify the results obtained, specific metrics were used to measure the quality, accuracy, and speed of the system's detection. The experiments were evaluated using the following metrics.

### 3.1. Metrics

To evaluate the detection performance, the following metrics were used according to the confusion matrix expressed in Equation (11). The used metrics were as follows:

$$ConfMtr = \begin{bmatrix} \text{True Positive} & \text{False Negative} \\ \text{False Positive} & \text{True Negative} \end{bmatrix} = \begin{bmatrix} TP & FN \\ FP & TN \end{bmatrix} \quad (11)$$

Precision, as expressed in Equation (12), was the measure of the precision of the bounding boxes, with their prediction accuracy being taken into account.

$$P = \frac{TP}{TP + FP} \quad (12)$$

Recall, as expressed in Equation (13), was how many objects were selected in the bounding boxes and could provide insight into the efficiency of the system during object detection of the class interest.

$$R = \frac{TP}{TP + FN} \quad (13)$$

Specificity, as expressed in Equation (14), was the rate of true negatives, i.e., the correct number of negative detections obtained on YOLO.

$$S = \frac{TN}{TN + FN} \quad (14)$$

F1-Score, as expressed in Equation (15), was a combination of the evaluation of the recall performance and the precision of the detection.

$$F1 = 2 \times \frac{P \times R}{P + R} \qquad (15)$$

### 3.2. Database

The proposed database consists of 3 sets of BD plant images as shown in Table 1.

**Table 1.** Database. Train and Test contain images of BD grade 3 to 5 plants from 5 to 22 years old. Scope contains images of grade 3 to 4 plants from plants less than 4 years old.

| Set | Size | Format | Quality |
|---|---|---|---|
| Train | 2000 | JPG | 5 Mpx. |
| Test | 400 | JPG | 8 Mpx. |
| Scope | 400 | JPG | 64 Mpx. |

The model training architecture was based on an intel Xeon CPU with 13 GB of RAM and an Nvidia Tesla T4 GPU. The dataset used consists of 2000 images of plants with BD, 90% of which was used for training and 10% for validation.

For the training process, we used the Train dataset, which contains the ground truth labels of the object, i.e., the reference points (*RP*) of the object within the image. To generate the RP of the object of interest (BD), it was necessary to manually label the images of the Train dataset using the image annotation tool LabelImg [31]. *RP* were created by an agricultural engineer from the Zitihualt company in order to have accurate annotations of the object in the image supported by an expert. Their expression can be seen in Equation (16).

$$RP = [ID, x, y, w, h] \qquad (16)$$

The *ID* parameter represents the object's class. The second and third parameters, *x* and *y* are, respectively, the *x* and *y* coordinate of the center bounding box, normalized by width and height of image. The fourth and fifth parameters, *w* and *h*, are, respectively, the width and height coordinates of the bounding box, normalized by image width and height [22,29].

### 3.3. Experiment 1: Detection Threshold Tuning

After training, we performed a YOLO tuning process according to parameters such as input image size and detection threshold. The tuning was used in JN to find an optimal configuration during BD detection. Figure 8 shows the results obtained according to the proposed metrics at 10 detection thresholds with a 416 × 416 input image.

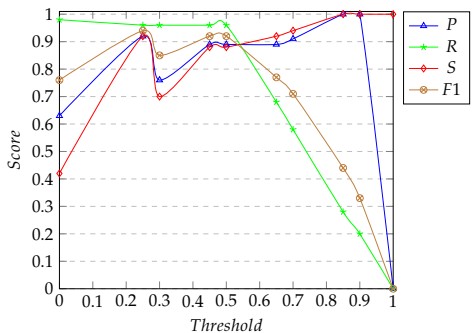

**Figure 8.** Comparison of detection thresholds. P—precision; R—recall; S—specificity; F1—F1-Score.

As shown in Figure 8, the detection threshold that obtained the best results was 0.25, with a significant synchrony in all metrics used, reaching scores above 90%. Although the 0.5 detection threshold provided an exact correlation of 8.9 in all metrics, at the time of displaying the detection, it generated an inference by detecting the same object in 2 different bounding boxes; for this reason, we decided to use the 0.25 threshold.

### 3.4. Experiment 2: Validation on the Test Dataset

For this experiment, the Test dataset was used, which consisted of 200 photographs of plants older than 5 years with BD incidences of grade 3 according to the CENIPALMA severity scale [19] and 200 photographs of plants older than 5 years without any incidence. According to the detection threshold set in the YOLO tuning process, the following confusion matrix Equation (17) was obtained.

$$ConfMtr = \begin{bmatrix} 192 & 8 \\ 16 & 184 \end{bmatrix} \tag{17}$$

The results obtained according to different literature metrics were 92% in precision, 96% in recall, 92% in specificity, and 94% in F1-score, according to the values in Equation (17). Figure 9 shows the ROC curve [32] obtained, from which an AUC of 0.94 was calculated.

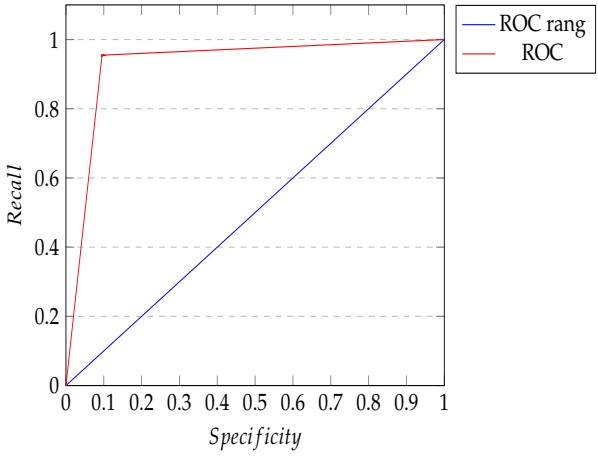

**Figure 9.** ROC curve obtained during validation on the Test dataset.

### 3.5. Experiment 3: Real-Time BD Detection

In this experiment, 20 hectares of 22-year-old oil palm plantations were evaluated. The average height of the crop was 23 to 26 m. Figure 10 shows two oil palm plants from one of the plots, showing the difference between a healthy plant and one with BD.

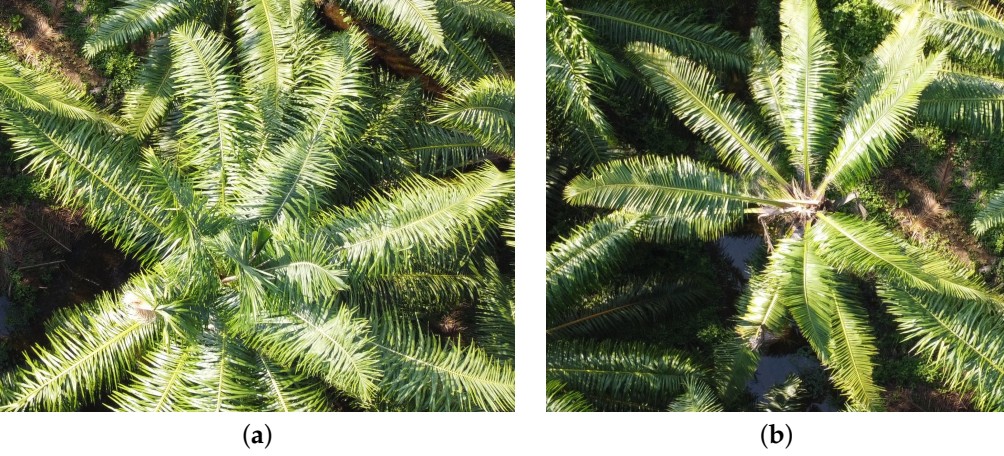

(**a**)  (**b**)

**Figure 10.** Oil palm plants 22 years old. (**a**) Healthy palm plant. (**b**) Palm plant with BD.

The evaluation of the detection in real time was carried out in the proposed system during 6 months of the year in order to evaluate its feasibility under different conditions that could occur in the crop; moreover, the following additional metrics were also taken into account:

FN and TP prediction: Figure 11a, shows how the model detects a plant with BD as a FN, but this case is repeated only in the first frames of the video due to the detection speed provided by the JN GPU, and 2 s later it is corrected to detect the same plant as the TP, as seen in Figure 11b.

TN and FP prediction: During crop analysis, YOLO keeps the FP number low because the camera angle and focal length are optimal for the detection work, while the TN number is kept almost accurate.

Visual range impact: This is the effect of the camera's field of view on the target. BD detection is performed at a maximum distance of 2 to 3 m above the plant bud; Figure 12 shows how the impact of field of view does not conflict with the detection of crop occurrences.

Impact of occlusion: An important feature of real-time detection is the ability to detect objects that appear partially in an image. In this case, this is a minor factor, as the field of view during the crop flyover provides a full view of the plant in question, and the crops are planted in batches of plants of the same age and therefore of the same height, which does not create overlap of one plant over another. Figure 13 shows how YOLO is able to detect a plant with BD that appears partially in the field of view of the system's camera.

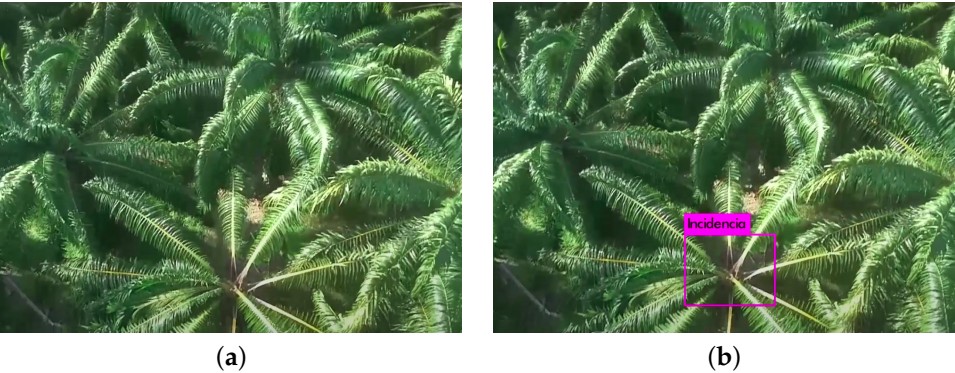

(**a**) (**b**)

**Figure 11.** Detection of palms with BD. (**a**) False negative. (**b**) True Positive.

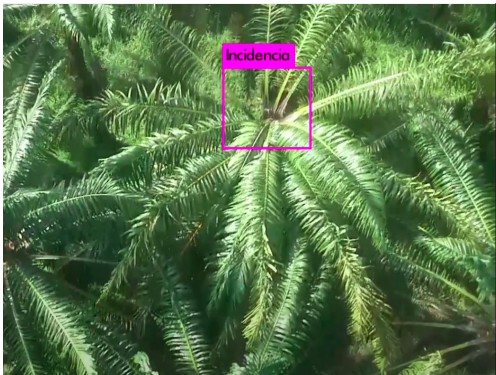

**Figure 12.** Visual range impact on crop.

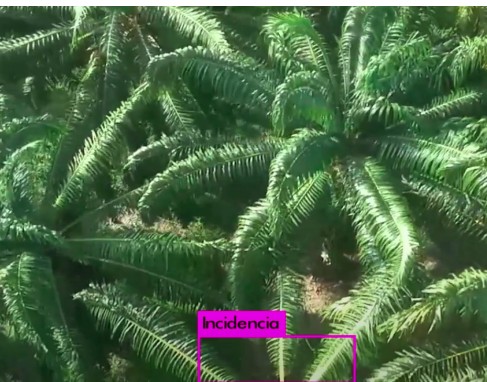

**Figure 13.** Impact of occlusions on crop.

Real-time detection speed: An important factor to take into account is the frames per second (FPS) at which the system manages to execute the algorithm since this influences the flight speed that the drone must have while flying over the crop. Figure 14 shows how the system manages to reach an average speed of 11.3 FPS, and although this is a low speed, more accurate detections are achieved. As a result, the system takes about 25 min to fly over one hectare of oil palm crop.

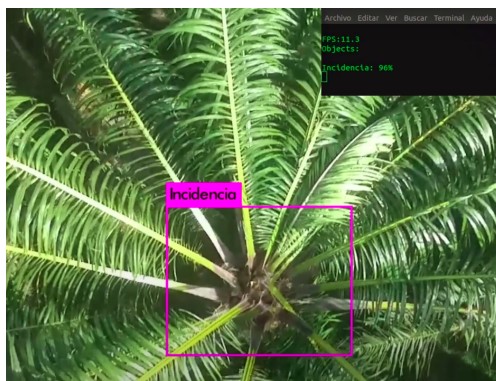

**Figure 14.** Real-time palm detection with BD.

*3.6. Experiment 4: Comparison with Current Methodology*

According to CENIPALMA, there is no formal methodology for detecting BD. Several studies, such as [19,24,33], indicate that for better management, it is essential to constantly monitor the crop for indicators such as dry leaves and/or abnormal bud growth. Companies such as Zitihualt, with the support of FEMEXPALMA [34], propose monitoring the entire crop for disease indicators. However, as mentioned above, this technique is not very reliable and requires too much physical effort on the part of the personnel in charge. For this experiment, the performance of the system was evaluated in 8 hectares of oil palm plantations, with the methodology currently used and the proposed methodology being compared.

For crop selection, certain characteristics were taken into account, such as the state of the crop, i.e., whether there are weeds or pruning in the plant, a characteristic that influences the detection of BD by the farmer, and the age of the crop, a characteristic directly related to the height of the plant. Similarly, the time required to evaluate one hectare of crop was recorded, which is an important factor when comparing the two methods. Table 2 shows the results obtained with both methods.

**Table 2.** Methodology Comparison. Both methods were evaluated using the precision, recall, and F1-score metrics to compare the method prediction and correct object detection using a confusion matrix.

| Condition of the Crop | Age of Crop | Detection Time | | Precision | | Recall | | F1-Score | |
|---|---|---|---|---|---|---|---|---|---|
| | | Current | Proposed | Current | Proposed | Current | Proposed | Current | Proposed |
| Clean crop | 5 | 2:20 | 0:25 | 86% | 100% | 86% | 86% | 67% | 92% |
| Dirty crop | 5 | 3:18 | 0:23 | 75% | 83% | 60% | 100% | 67% | 89% |
| Clean crop | 9 | 2:50 | 0:24 | 100% | 75% | 67% | 100% | 80% | 100% |
| Dirty crop | 9 | 3:15 | 0:15 | 50% | 80% | 25% | 100% | 33% | 89% |
| Clean crop | 15 | 3:00 | 0:23 | 100% | 67% | 67% | 100% | 80% | 100% |
| Dirty crop | 15 | 3:30 | 0:30 | 50% | 100% | 67% | 67% | 57% | 100% |
| Clean crop | 22 | 3:10 | 0:26 | 35% | 100% | 50% | 100% | 33% | 100% |
| Dirty crop | 22 | 3:35 | 0:30 | 67% | 100% | 40% | 80% | 40% | 89% |

According to Table 2, it can be concluded that during the evaluation of crops with the proposed methodology, there is a higher efficiency in terms of accuracy and execution time with respect to the currently used methodology. The results of the comparison show that the current methodology tends to present difficulties at the time of BD detection when there are weeds in the crop and the size of the plant exceeds 10 meters; this is in contrast with the proposed methodology, which maintains its effectiveness and achieves 100% effectiveness in some crops and is superior in terms of detection time, reducing the time required by 75%.

### 3.7. Experiment 5: Validation on the Scope Dataset

For this experiment, the Scope dataset was used, which consists of 200 images of plants with completely different environments on which YOLO was trained. The following results were obtained.

Detection of BD in plants younger than 4 years and with BD grade 3 resulted in too many FNs, as shown in Figure 15a. However, when a close-up of the plant bud was performed, the model was able to detect a diseased plant as a TP, as shown in Figure 15b, but the accuracy was lost because YOLO confuses healthy plants with diseased ones, an example of which is the plant in Figure 15c, which shows an FP.

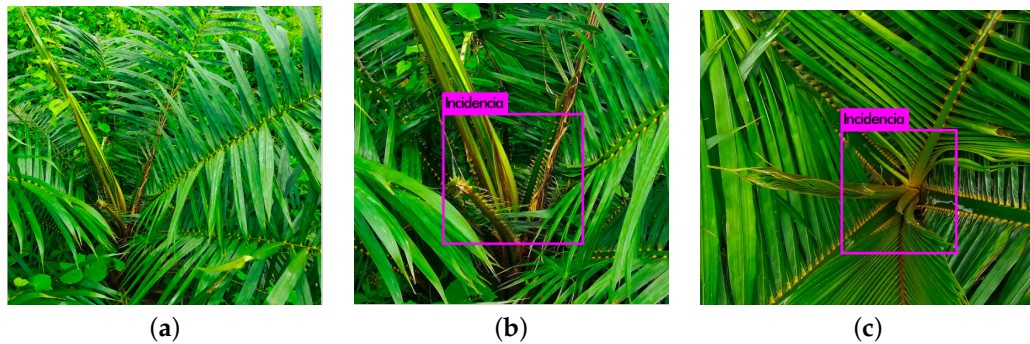

**Figure 15.** Detection of 4-year-old palms with BD. (**a**) False negative in a plant with grade 3 BD. (**b**) True positive in a plant with grade 3 BD. (**c**) False positive in a healthy plant.

For plants with BD grade 4 during the evaluation, the model obtained a precision of 43% and a recall of 68%, although these are low values considering that YOLO was evaluated with completely different plants in terms of training dataset characteristics. It can be concluded that, in general, YOLO could be optimal for detection work when the training data are limited with a certain degree of accuracy.

## 4. Conclusions

The methodology proposed in this study represents a significant improvement in the detection of BD in oil palm crops, offering higher accuracy and faster response time than those of the current method. The use of a drone equipped with a Jetson Nano system reduces the time required to monitor a hectare of crop by 75%, with a 40% increase in accuracy compared to the current method. In addition, the method allows BD to be detected in its early stages, enabling effective disease control and reducing the amount of crop protection products required.

The results obtained during the evaluation of the algorithm on the Test dataset show an precision of 92% and a recall of 95%, reaching an AUC of 0.94 on the ROC curve. These results demonstrate that the proposed methodology provides an effective and reliable means of detecting BD in oil palm crops, which can prevent the spread of pests and improve crop yields. During real-time detection, the system demonstrates robustness by being able to detect plants with BD incidences with an precision of 92% and a recall of 96%, indicating a high capacity to detect the disease. The same is true for the detection of plants with BD that were not part of the dataset used to train the model. The results showed that the proposed system was able to detect with an precision of 43% and recall of 68%, suggesting that the model is quite robust in detecting plants with BD in real situations where plants not previously seen in the training set can be found. However, it is important to note that this test was performed with a limited number of plants, and further testing is needed to validate the robustness of the system in different situations.

The application of this technology provides a more efficiency, efficient, and sustainable tool for disease detection in oil palm plantations, reducing the excessive use of chemicals and improving crop yields. It also provides the ability to collect and analyze long-term data to study the evolution of bud degeneration and improve disease management. In addition, the proposed technology has great potential for application in other areas of agriculture, contributing to the development of more sustainable and efficient practices.

As it could be observed, the proposed system presents feasible results in the detection of BD; however, it presents some limitations, which future is being planned to solve. Among these issues is the dependence on weather conditions, since the system is based on capturing images from a drone, so it can be affected by adverse conditions such as rain, fog, or low light that limits visibility. Therefore, we intend to improve both the flying equipment and the camera used.

Another issue is the sensitivity to changes in the degree of BD and the age of the plant: the system is trained to detect certain characteristics of a type of plant, so if this is different from the one that has been trained, the system may have problems detecting the degeneration of the bud. Thus, we intend to increase the set of training data to increase the detection capability of the algorithm to new data.

**Author Contributions:** Conceptualization, A.V.-R., M.M.-C., D.M.-V. and A.L.-Á.; methodology, A.V.-R., M.M.-C., D.M.-V. and A.L.-Á.; software, A.V.-R., M.M.-C., D.M.-V. and J.d.J.R.; validation, A.V.-R., A.L.-Á. and M.M.-C.; formal analysis, A.V.-R., D.M.-V., J.d.J.R. and M.M.-C.; investigation, A.V.-R., M.M.-C., D.M.-V., J.d.J.R. and A.L.-Á.; resources, A.V.-R., D.M.-V., A.L.-Á., M.M.-C. and J.d.J.R.; writing—original draft A.V.-R. and M.M.-C.; writing—review and editing, A.V.-R., A.L.-Á., M.M.-C., D.M.-V. and J.d.J.R.; visualization, A.V.-R. and M.M.-C.; supervision, A.V.-R., D.M.-V., M.M.-C., A.L.-Á. and J.d.J.R. All authors have read and agreed to the published version of the manuscript.

**Funding:** This research received no external funding.

**Institutional Review Board Statement:** Not applicable.

**Informed Consent Statement:** Not applicable.

**Data Availability Statement:** Not applicable.

**Acknowledgments:** The authors would like to thank CONAHCYT, Tecnológico Nacional de México/Centro Nacional de Investigación y Desarrollo Tecnológico and Zitihualt S.P.R. de R.L. for their support throughout this work.

**Conflicts of Interest:** The authors declare no conflict of interest.

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
