# Peer review of "Real-Time Detection of Bud Degeneration in Oil Palms Using an Unmanned Aerial Vehicle"

_2673-4117, doi:10.3390/eng4020090_

Round 1

Reviewer 1 Report

The authors present Real-time Detections of Bud Degeneration in Oil Palms using an Unmanned Aerial Vehicle. The study is interesting.  In general, the main conclusions presented in the paper are supported by the figures and supporting text. However, to meet the journal quality standards, the following comments need to be addressed.

•           Abstract: Should be improved and extended. The authors talk lot about the problem formulation, but novelty of the proposed model is missing. Also provided the general applicability of their model. Please be specific what are the main quantitative results to attract general audiences.

•           The introduction can be improved. The authors should focus on extending the novelty of the current study. Emphasize should be given in improvement of the  model (in quantitative  sense)  compared to   existing  state-of-the art models.

•           More details about network architecture and complexity of the model should be provided.

•           what about comparison of the result with current state-of-the art models?  Did authors perform ablation study to compare with different models?

•           What are the baseline models and benchmark results? The authors may compared the result with existing models evaluated with datasets

•           Conclusion parts needs to be strengthened.

•           Please provide a fair weakness and limitation of the model, and how it can be improved.

•           Typographical errors: There are several minor grammatical errors and incorrect sentence structures. Please run this through a spell checker.

 Discussions of relevant literature could be further enhanced, which can help better motivate the current study and link to the existing work. Authors might consider the following relevant recent work in the field of applying machine learning techniques to better motivate the usefulness of machine learning approaches, such as

 see : - Drones 2023, 7(2), 81; https://doi.org/10.3390/drones7020081 ;

- Neural Networks 2022 https://doi.org/10.1016/j.neunet.2022.05.024

-arXiv (2023)  https://doi.org/10.48550/arXiv.2303.04275

Hence they should be briefly discussed in the related work section.

Needs some minor revisions 

Author Response

The attached document details the comments addressed.

Reviewer 2 Report

The paper presents a real-time detection of bud degeneration in oil palms using an unmanned aerial vehicle. The paper is quite organized, although the scientific contribution is less visible. Here is my comments:

1. The paper doesn't explicitly describe which YOLO version was used for the experiment. There are several YOLO versions with slightly different configuration but big discrepancy in their performance. 

2. In lines 145-154, the authors describe the basic mechanism of YOLO. Did the authors modify the structure of original YOLO such that they could do a transfer learning? If the authors didn't do any adaptation to the original YOLO structure (and its parameters) for accommodating a transfer learning, how can the authors make sure that YOLO can give a good prediction over BD?

3. Since YOLO is a CNN model with pre-trained data, how do the authors incorporate their own data into YOLO? Did the authors consider the existing YOLO parameters regardless of how they trained the model? Also, standard YOLO training procedure usually involves providing labelled data. The authors describes that the training data contains labelled object in the form of Reference Points (RP). However, the authors didn't explain how the RPs are collected or generated. Did the authors manually create bounding boxes in the training data or using some tools for automatic annotation? Please describe in detail about the dataset generation for training YOLO!

4. To give readers insightful idea of how BD looks like in various palm tree conditions, it is better if the authors give several examples of BD images (from different incidence grades) as well as several examples of normal (non-BD) images. From Figure 9 to 12, it is still difficult even for human reader to distinguish normal palm trees from trees with BD! This is useful in order to understand how YOLO might bevahe on these condition. 

5. In Figure 7, please use English labels for the graph axes! Figure 8 should have axes labels to make it clear and more understandable for readers not familiar with ROC!

6. Section 3.6 describes and experiment for comparison with current methodology. However, it is not clear, what is that current methodology. Does it use certain algorithm or different measurement systems? Please describe in detail!

7. In general, the English grammar, especially used in several figure/table captions could be improved. They are too short and less informative. For example in Figure 1, rather than "Methodology proposed", it should be written as "Proposed methodology". Even better, that caption should be written longer, explaining the detailed information about the figure. This can also be applied to the other figure captions as well. In general, figure captions in the this paper are too short and less informative. They should be made longer and give a detailed information on certain/specific important aspect in the figure.

In general, the English grammar is OK, but the captions for figure/table could be improved. Again, they are too short and less informative. The captions should be written longer, explaining the detailed information about the figure/table. They should clearly explain/describe certain/specific important aspect in that figure/table.

Author Response

(The authors gave the same response as above.)

Round 2

Reviewer 1 Report

No further comments

Reviewer 2 Report

The authors have properly addressed issues mentioned in the previous comments. Some insightful additional contents as well as relevant references have been added to the manuscript.